# The Design of Ternary Composite Polyurethane Membranes with an Enhanced Photocatalytic Degradation Potential for the Removal of Anionic Dyes

**DOI:** 10.3390/membranes12060630

**Published:** 2022-06-17

**Authors:** Usman Zubair, Muhammad Zahid, Nimra Nadeem, Kainat Ghazal, Huda S. AlSalem, Mona S. Binkadem, Soha T. Al-Goul, Zulfiqar Ahmad Rehan

**Affiliations:** 1Department of Textile Engineering, School of Engineering and Technology, National Textile University, Faisalabad 37610, Pakistan; usman.zubair@ntu.edu.pk (U.Z.); nimranadeem692@gmail.com (N.N.); 2Department of Chemistry, University of Agriculture Faisalabad, Faisalabad 38040, Pakistan; zahid595@gmail.com; 3Department of Materials, School of Engineering and Technology, National Textile University, Faisalabad 37610, Pakistan; ghazalakbar999@gmail.com; 4Department of Chemistry, College of Science, Princess Nourah bint Abdulrahman University, Riyadh 11671, Saudi Arabia; husalsalem@pnu.edu.sa; 5Department of Chemistry, College of Science, University of Jeddah, Jeddah 21589, Saudi Arabia; 04100507@uj.edu.sa; 6Department of Chemistry, College of Sciences & Arts, King Abdulaziz University, Rabigh 25732, Saudi Arabia; salgoul@kau.edu.sa

**Keywords:** antifouling membranes, photocatalysis, aminated graphene oxide, ternary composite, wastewater treatment, Ni

## Abstract

Photocatalysis is an efficient and an eco-friendly way to eliminate organic pollutants from wastewater and filtration media. The major dilemma coupled with conventional membrane technology in wastewater remediation is fouling. In this study, the photocatalytic degradation potential of novel thermoplastic polyurethane (TPU) based NiO on aminated graphene oxide (NH_2_-GO) nanocomposite membranes was explored. The fabrication of TPU-NiO/NH_2_-GO membranes was achieved by the phase inversion method and analyzed for their performances. The membranes were effectively characterized in terms of surface morphology, functional group, and crystalline phase identification, using scanning electron microscopy, Fourier transformed infrared spectroscopy, and X-ray diffraction analysis, respectively. The prepared materials were investigated in terms of photocatalytic degradation potential against selected pollutants. Approximately 94% dye removal efficiency was observed under optimized conditions (i.e., reaction time = 180 min, pH 3–4, photocatalyst dose = 80 mg/100 mL, and oxidant dose = 10 mM). The optimized membranes possessed effective pure water flux and excellent dye rejection (approximately 94%) under 4 bar pressure. The nickel leaching in the treated wastewater sample was determined using inductively coupled plasma-optical emission spectrometry (ICP-OES). The obtained data was kinetically analyzed using first- and second-order reaction kinetic models. A first-order kinetic study was suited for the present study. Besides, the proposed membranes provided excellent photocatalytic ability up to six reusability cycles. The combination of TPU and NH_2_-GO provided effective strength to membranes and the immobilization of NiO nanoparticles improved the photocatalytic behavior.

## 1. Introduction

The impact of rapid industrialization on the quality of water sources is undeniable. The textile and the printing industries are among the major industries contributing to the release of huge amounts of wastewater containing azo dyes and aromatic pollutants [1]. As per an estimation, approximately 10,000 dyes are available in the market with an annual production of 7 × $10^{5}$ metric tons [2,3,4,5,6]. It takes approximately 100 L of water to dye 1 kg of fabric. Therefore, a huge amount of dyeing material is discharged directly into the environment annually. These pollutants possess a high resistance to degradation. The carcinogenic and the mutagenic effects of these pollutants in water may cause liver and kidney cancer, abnormal functioning of the reproductive system, nervous disorders, and many other diseases [7]. Therefore, effective and advanced wastewater treatment technologies with a non-selective oxidation/degradation of inorganic–organic pollutants are direly needed in the current scenario.

Membrane technology has emerged as a potential technology for wastewater remediation due to its fascinating physicochemical properties, such as eco-friendliness, a compact modular structure, low chemical sludge production, facile maintenance, excellent separation capability, and a high quality of treated water [8,9,10]. Numerous polymers are available in the market to fabricate membranes with desired properties [11,12,13,14,15]. The well-documented membrane technology for wastewater treatment includes reverse osmosis (RO), nanofiltration (NF), ultrafiltration (UF), and microfiltration (MF) membranes classified on the basis of increased pore size for the rejection of various sized substances [16,17]. The porous UF and MF membranes are usually fabricated by the phase inversion method which involves controlled precipitation of polymers into hollow fibers or thin films to fabricate a porous membrane structure [10]. In comparison to other membranes, the thin film composite (TFC) membranes have the characteristics of being a highly permeable, exceptionally selective (due to a dense, ultra-thin layering by interfacial polymerization), and fast-growing family of membranes [18]. Thermoplastic polyurethane (TPU) is a block copolymer with extensive properties based on its soft and its hard segment configurations [8]. The exceptional properties of TPU make it a suitable candidate for various applications, including wastewater remediation. However, membrane fouling is still a real bottleneck in the practical application of this technology. During water treatment, the major foulants arise from hydrocarbons, humid substances, macromolecular organic matter, mineral scaling, and suspended sludge [10]. Antifouling can be achieved by a fouling release mechanism, a fouling resistant mechanism, and a fouling attacking mechanism [10]. Therefore, modifications of membranes are gaining attention to enhance and to improve the antifouling ability of membranes. Among others, graphene oxide is an emerging potential candidate that can impart surface hydrophilicity, and it can also serve the function of being a nanofiller in a membrane matrix [19].

Graphene oxide (GO) is a graphene derivative with extensive and well-distributed oxygen-containing functionalities, such as hydroxyl, epoxy, carboxylic, and carboxyl groups, responsible for its hydrophilic and antibacterial character [20]. GO exhibits excellent chemical and mechanical stability and a large specific surface area [20,21,22]. Owing to these exceptional features, GO is currently being exploited as a superficial filler in polymeric membranes [23]. Due to its thinner structure, robust chemical and mechanical action, and effective ion-selective properties, graphene oxide holds the advantage of being a new generation material in wastewater treatment [24,25,26]. Various studies in the multidimensional field reported the effective role of GO in a polymeric membrane composite, including in water remediation technology [27,28,29]. GO possesses easy functionalization capability, a planer structure, a high aspect ratio, and strength. However, compositing the GO with membrane technology results in a weak positive charge density of membranes because of the hydrolysis of COOH groups present in GO [30]. The chemical functionalization of GO can significantly expand its applications by overcoming this issue. For example, Xu and coworkers reported the grafting of amine groups (-NH_2_) at the edges of graphene quantum dots (GQDs) to produce GQDs-NH_2_. This can serve as an aqueous monomer to incorporate -NH_2_ and -OH functionalities during interfacial polymerizations [31]. Effective interfacial polymerization improves the physicochemical properties of membranes (chargeability, hydrophilicity, morphology, etc.).

Engineered heterostructure semiconductor photocatalysis is the emerging domain in the area of science and technology, where the advantages of multicomponents can be achieved in a single composite [32]. The salient features of this technology involve the utilization of metal oxide or metal-based nanoparticles and being non-toxic and cheap in cost, with a high chemical stability and effective reusability characteristics. Loading of nanomaterials into the polymeric membrane structure is another emerging strategy to improve the membrane’s performance [33]. Besides other metal oxide-based photocatalytic nanomaterials (such as Fe_2_O_3_, TiO_2_, ZnO, etc.) [34,35], the NiO NPs are the potential candidate, exhibiting a wide range of contaminant removal applications with an energy bandgap of 3.4 eV [36,37]. Under irradiation of a suitable wavelength, the excitation of charge carriers prompts the photodegradation of various pollutants [38]. The potential applications of NiO NPs have been explored in the field of sensors [39], photocatalysis [40], space technology [41], cosmetics [42], magnetism, energy, and catalysis [43,44], etc. The immobilization of NiO NPs on GO could result in improved physicochemical characteristics [45]. The functional moiety on GO helps in successful chemical interactions between NPs and the GO matrix.

So, here we proposed the combination of a polymeric TPU membrane with an NiO/NH_2_-GO composite for an improved photocatalytic degradation mechanism. The novel composite was supposed to exhibit an effective implementation in antifouling membrane technology for wastewater treatment as up to 80% efficiency was achieved even after six reusability cycles. Successful synthesis was achieved and confirmed by key analysis techniques (i.e., XRD, FTIR, SEM). The potential of prepared membranes were analyzed against MO (as model pollutant) dye and up to 94% degradation was achieved under optimized reaction conditions. The nickel leaching in the treated wastewater sample was determined using inductively coupled plasma-optical emission spectrometry (ICP-OES). Two kinetic models, namely first- and second-order kinetic models, were applied to estimate the reaction kinetics. The membrane’s porosity, pure water flux, and dye rejection experiments were also performed. As-fabricated composite membranes offer improved antifouling behavior, owing to the combined effect of TPU with NiO nanoparticles intercalated within aminated graphene oxide (NH_2_-GO).

## 2. Materials and Methods

### 2.1. Chemicals

The chemicals used for the present research were all analytical grade and used as obtained. Graphite powder (<20 [46] synthetic), hydrogen peroxide (H_2_O_2_: 30% *w*/*w* in H_2_O), potassium permanganate (KMnO_4_: ACS reagent grade), ammonia solution (NH_4_OH: 25% for analysis EMSURE ISO, Reag. Ph Eur.), sulphuric acid (H_2_SO_4_: Assay 95%), thermoplastic polyurethane, nickel nitrate hexahydrate (Ni(NO_3_)_2_.6H_2_O: trace metal basis, 99.99%), and sodium nitrate (ACS grade, ≥99.0%) were purchased from SIGMA ALDRICH. Whereas sodium hydroxide (NaOH, 98%), dimethylformamide (Assay 99.9%, SGR grade), ethylene glycol (Assay 99%, FP 116 °C), phosphoric acid (H_3_PO_4_: 85% HPLC grade), and hydrochloric acid (HCl: assay 36%) were obtained from DAEJUNG. Deionized water was used for the preparation of all solutions.

### 2.2. Synthesis

#### 2.2.1. Synthesis of GO

A modified Hummers method was selected for the fabrication of GO. To do so, 1.0 g of powder graphite and 0.5 g NaNO_3_ were added to 25 mL of sulfuric acid under continuous mechanical stirring in an ice bath. The reaction temperature was maintained below 10 °C. Afterward, 5.5 g KMnO_4_ was added within 0.5 h under constant stirring. The as-prepared suspension was continuously stirred for many hours at the reaction temperature of 30–35 °C. Then, a dropwise addition of 50 mL of deionized water resulted in a high exothermic reaction, raising the temperature of the system above 90 °C. This temperature was maintained for 10 min by hot plate. After this, 120 mL of deionized water was added, followed by the addition of 10% hydrogen peroxide. Graphitic oxide was obtained in the form of a brownish yellow paste. The slurry was rinsed with a 5% solution of HCl, ethanol, and deionized water, respectively, many times until neutral. The graphitic oxide was ultrasonicated for a specific time to obtain a stable GO. The obtained GO was then air dried at 60 °C.

#### 2.2.2. Synthesis of Aminated Graphene Oxide (NH_2_-GO)

NH_2_-GO was prepared by a solvothermal route using GO with ammonia and EG. Firstly, 1 g of GO in 25 mL DI water was ultrasonically dispersed afterward and 210 mL ethylene glycol was added to the reaction. Then, 6 mL of ammonia water was added. This reaction mixture was then added to a Teflon lined autoclave reactor for the chemical reaction under a high pressure and temperature for 15 h. After cooling to room temperature, the aminated GO was taken out, rinsed with DI water until neutral, and air dried at 60 °C [9,10].

#### 2.2.3. Synthesis of Nickel Oxide NiO Nanoparticles

NiO NPs were fabricated by the sol-gel technique. In a typical process, 0.03 M nickel nitrate hexahydrate was added to 60 mL of deionized water. The solution was stirred magnetically at 75 °C and 300 rpm. Then, the solution pH was maintained at approximately 11. The solution turned green. To become diphasic, the solution was left aside for a specific time. The green gel was obtained after filtration, which was dried under a specific temperature and ground to form a powder [11].

#### 2.2.4. Fabrication of NiO/NH_2_-GO Composite

NiO/NH_2_-GO was prepared by compositing NiO nanoparticles and NH_2_-GO under a hydrothermal treatment. Typically, NiO and NH_2_-GO were dispersed in 30 mL deionized water with three weight ratios (i.e., 1:1, 2:1, and 1:2). The suspensions were ultrasonicated for 30 min and transferred into a 50 mL capacity autoclave reactor. The reactor was placed in oven for 12 h at 180 °C. After the temperature of the reactor dropped down near to room temperature, the samples were taken out and rinsed with DI water and ethanol and air dried at 60 °C for 6 h.

#### 2.2.5. Fabrication of NiO/NH_2_-GO-Based Thermoplastic Polyurethane Membrane (TPU- NiO/NH_2_-GO)

The composite membranes, i.e., TPU-NiO/NH_2_-GO were fabricated by the phase inversion method. TPU (18%) with variable concentrations of NiO/NH_2_-GO nanocomposite were added to Dimethylformamide for the preparation of the casting solution. The casting solution was stirred for 4–6 h to homogenize it. Then, the solution was left under stirring for approximately 24 h. Afterward, the solution was casted over a glass sheet using a casting knife. After 90 s, the casted membrane was transferred in a coagulation bath for approximately 2–3 min. When the membrane was detached from the glass surface, it was rinsed with DI and dried at room temperature. The composition of different thermoplastic polyurethane membranes is presented below in Table 1.

### 2.3. Characterizations

Synthesized TPU-NiO/NH_2_-GO membranes were characterized by some analytical techniques. The surface morphology of membranes was characterized using SEM (ISB, TESCAN, SEM HV = 10.0 kV). The identification of functional moieties in TPU- NiO/NH_2_-GO membranes were analyzed by FTIR (Agilent Technology Cary 360 FTIR spectrophotometer) and the structure and the crystallinity of composite membranes was evaluated by XRD (Philips PANalytical Xpert pro DY 3805 powder XRD). The inductively coupled plasma-optical emission spectrometry (ICP-OES Agilent 5110) was performed to analyze the leaching of Ni metal ions into a treated wastewater sample. Additionally, the water permeability of the membranes was also checked for performance evaluation on lab-developed filtration assembly. Membrane porosity was determined using a gravimetric method [47]. Considering this method, the membrane’s average porosity can be determined in terms of overall void fraction, which can be calculated as the pore volume divided by the total membrane volume. All the membranes were oven dried and precisely weighed. The weighed membranes were then immersed into the kerosene oil for 24 h and reweighed. The average porosity was calculated using the formula as below:(1)$$\varepsilon_{m}\;\%=\left[\frac{W_{1}-W_{2}}{D_{k}}\right]/\left[\frac{W_{1}-W_{2}}{D_{k}}+\frac{W_{2}}{D_{pol}}\right]\times 100$$
where
$$W_{1}=weight\;of\;wet\;membrnae,W_{2}=weight\;of\;dry\;membrane,$$
$$D_{k}=denisty\;of\;kerosene\;oil\;\left(0.82\frac{g}{cm^{3}}\right),\;D_{pol}=density\;of\;polymer$$

### 2.4. Photocatalytic Experimentation

The photocatalytic potential of all catalysts NiO, NH_2_-GO, NiO/NH_2_-GO (2:1), NiO/NH_2_-GO (1:1), and NiO/NH_2_-GO (1:2) were analyzed against MO dye degradation. After this, the best combination of NiO/NH_2_-GO was selected for making a composite with TPU, having variable ratios for further optimization. All the reactions were conducted in ultraviolet irradiation in a digital chamber (UV model:ZM144W) equipped with six ultraviolet lamps, each with energy of 18 W. The radiometer (UVX digital, Analytic Jena, probe UVX-25 for 254 nm) was used to measure the light intensity of UV irradiation. The optimization (within specific range) of influential parameters, i.e., pH (2–9), oxidant dose (0–16 mM), and reaction time (30–180 min), was achieved successively. Generally, the 10 ppm/100 mL dye solution of MO was chosen for the degradation study. After that, a particular amount of catalyst was added into the respective beaker after the maintenance of the pH of the solution. After sonication (to make effective distribution of particles in dye) the samples were placed in the dark for 30 min under constant stirring to achieve adsorption equilibrium. Finally, the samples were placed in a UV chamber for photocatalysis of the MO dye, and samples were taken out after successive intervals. The absorbances of aliquots were recorded at λ_max_ = 465 nm for MO using a UV-Visible spectrophotometer (CECIL CE 7200) after separation of catalysts and dye degradation percentage was calculated using the formula below:$$Degradation\;\left(\%\right)=\left[1-Cf/Co\right]*100$$
where $C_{o}\;$is the initial concentration of dye and $C_{f}\;$is the dye concentration after a different treatment time.

## 3. Results and Discussion

### 3.1. Physicochemical Characterization

#### 3.1.1. FTIR

The functional group identification in prepared materials was carried out via FTIR analysis, and the obtained results are presented in Figure 1. In the FTIR spectra of TPU with NiO and with all GO composite membranes, a strong peak around 600 cm^−1^ confirms the presence of NiO NPs [31]. The peaks corresponding to oxygen-containing functional groups include C-O (1080 cm^−1^), C-OH (1230 cm^−1^), C = O (1715 cm^−1^), and OH (3320 cm^−1^) all ascribed to the graphene oxide [32]. However, the strong peak at 3320 cm^−1^ in composite membranes is attributed to the OH group bonded to the N-H group in TPU [33,34]. Another thing to be considered is that the peak intensity corresponding to the N-H group in TPU based membranes increases with the increase in the concentration of aminated graphene oxide (Figure 1 inset). Therefore, at a higher concentration of aminated GO, effective bonding between OH groups and the N-H group was observed.

#### 3.1.2. XRD

Figure 2 presents the XRD analysis of all NiO/NH_2_-GO/TPU membranes. A broad peak at 11°–15° is ascribed to GO in membranes and a broad peak at 15°–23° denotes TPU. Slight variations in peak intensities have been observed in different composite membranes. The higher graphene oxide content results in a little reduction of the TPU peak intensity. However, in the best combination, i.e., TNG-08 the peak intensities are comparable. This could be the result of the restriction in the PU chain movement due to a good interaction between TPU and the GO matrix [35,36]. Therefore, TNG-08 instead of TNG-10 can be chosen as the best possible TPU based membrane combination for effective implementation in any field of study. Besides, crystal planes attributed to the NiO are present, and peak intensities seem to be suppressed due to various interactions with composite membranes. However, the strong peak at 2 theta values of 44° (200) confirms the presence of NiO NPs in all membranes [37].

#### 3.1.3. SEM

The morphology of selected samples (i.e., TN-02, TNG-02, and TNG-08) was analyzed by SEM imaging and presented in Figure 3. The SEM image of TPU with NiO shows that the polymer matrix is well decorated with NiO NPs, whereas the membranes containing aminated graphene oxide showed well scattered GO sheets throughout the membrane matrix. The rough distribution of NH_2_-GO onto a polymer matrix increases with increasing NH_2_-GO content (i.e., from TNG-02 to TNG-08). This also provides an improved adsorption site for pollutants followed by effective photocatalytic degradation of adsorbed pollutants [38].

#### 3.1.4. Mechanical Performance

Mechanical properties are of vital importance in filtration membranes. Even though such designed membranes can provide a high-water flux and pollutant rejection, they need to have good mechanical strength and durability to withstand operating pressures. Interestingly, TPU membranes exhibited higher tensile strength on the inclusion NiO/NH_2_-GO at greater quantities. This enhancement in tensile strength can be attributed to the interaction (H-bonding) of NH_2_/OH groups on GO with C = O/NH groups on TPU. The similar was revealed during the FTIR analysis of composite membranes. These interactions greatly contribute toward the intermolecular connections among the polymeric chains that foster the mechanical strength of membranes. The greater the interactions among the molecules, the higher the mechanical strength of the sample, as prescribed in Figure 4.

### 3.2. Influence of Key Reaction Parameters

The degradation of pollutants using photo catalytically active materials is a multifactor effecting chemical process. The key factors that affect the degradation process are described below.

#### 3.2.1. Effect of Solution pH

The pH of the solution is one of the influential parameters that affect the degradation of pollutants during the adsorption process. Figure 5 illustrates that the pH of the solution has a significant effect on MO degradation. The pH of the solution affects the surface functional groups of catalysts and the degree of ionization of contaminants. Therefore, the pH of the dye solution was varied from 2 to 9 by using 0.1 M NaOH and HCl solutions. The results showed that at an acidic pH, all the catalysts showed effective degradation. The effective photocatalytic degradation behavior under acidic conditions could be explained based on a point zero charge (pzc) on the surface of metal oxide intercalated aminated graphene oxide nanocomposites. It has been previously reported that the pHpzc on the surface of metal oxide intercalated aminated graphene oxide was 8.2 [39]. At a solution pH > pHpzc, the surface of the adsorbent becomes negatively charged, and at pH < pHpzc, the surface becomes positively charged [40]. The MO dye is an anionic pollutant, which favors an adsorption process under acidic conditions. It means at below 8.2 or near it, the adsorption of MO is a favorable process. The effective adsorption (between adsorbate (dye) and adsorbent (NiO/NH_2_-GO)) is mandatory for photocatalysis [41,42]. Therefore, pH 4 was the best solution environment for effective adsorption of dye molecules on a catalyst surface as best efficiency was achieved at pH 4 in the case of aminated GO based NiO nanocomposites. At a higher pH, besides the surface charge, another important factor that causes a reduction in degradation efficiency is the scavenging of OH radicals by OH ions (form base). The scavenging of potential radicals means less availability for dye degradation [43].

#### 3.2.2. Effect of Oxidant Dose

Hydrogen peroxide is a boosting agent in the dye photocatalytic degradation process. The hydroxyl radicals (generated from the dissociation of hydrogen peroxide) result in activation of the catalyst’s surface, which in turn increases the affinity of pollutant molecules with the catalyst surface [44]. Therefore, optimization of the oxidant dose was conducted in a predefined range (0–16 mM), keeping the pH, MO concentration (25 ppm), reaction time, and catalyst concentration constant. The obtained results are presented in Figure 6.

The results inferred a successive increase in the degradation of the MO dye, when the oxidant amount was increased. This direct relationship between dependent and independent variables is due to the availability of the effective hydroxyl radical when the concentration of hydrogen peroxide was increased. These are the results of the following reactions taking place in the reaction media [40]:(2)$$H_{2}O_{2}+O_{2}^{-\cdot }\to HO^{-}+˙OH+O_{2}$$
(3)$$H_{2}O_{2}+e_{\left(CB\right)}^{-}\to HO^{-}+˙OH\;$$

However, after the establishment of equilibrium between the oxidant dose and other operating parameters in the reaction, no effective increase in dye degradation was observed with the further increase in the oxidant dose [48,49]. The possible reason for this is the production of ${\mathrm{HO}}_{2}^{\cdot }$ (hydrogen per oxide radicals) due to following reactions:(4)$$H_{2}O_{2}+˙OH\to H_{2}O+HO_{2}^{\cdot }$$
(5)$$HO_{2}^{\cdot }+˙OH\to H_{2}O+O_{2}$$

${\mathrm{HO}}_{2}^{\cdot }$ radicals exhibit far less efficiency in the degradation process, and they also cause deactivation of hydroxyl radicals.

Considering the effectiveness of oxidants in the present study, 10 mM hydrogen per oxide was found to impart an efficient role in oxidative photocatalysis, as approximately 94% of dye degradation was achieved within 180 min of photocatalysis with a 10 mM H_2_O_2_ addition.

#### 3.2.3. Effect of Reaction Time

The effective implementation of a chemical process is strongly linked to the reaction time required to achieve desirable results. Therefore, the reaction time for photocatalytic dye degradation using composite membranes was optimized. Different time intervals, i.e., 20–180 min was selected for photocatalysis reaction. The aliquots were collected after sequential time gaps and percentage degradation was assessed for a specific time (Figure 7). The results showed that in the beginning, the photocatalytic dye degradation rate was higher, and it reached a maximum at approximately 180 min under UV irradiation. Besides, the photocatalytic potential of TPU alone is insignificant as compared to the composite membranes containing NiO/NH_2_-GO. The effective dye degradation was attributed to the enhanced charge immobilization in NiO NPs when aminated GO was incorporated. The best combination for the current study was found to be TNG-08, as shown in Figure 7. Figure 7b contains the MO dye with the TNG-08 membrane. The results showed successive removal of the MO dye with the passage of time. After 180 min of reaction time, almost 95% of the MO had been degraded under UV irradiation, with the optimized condition of other parameters.

#### 3.2.4. Stability of Composite Membrane (TNG-08) in Terms of Reusability

The advantage of using composite membranes in wastewater treatment is its effective reutilization due to antifouling behavior. The antifouling characteristics in the pristine membrane were achieved by utilizing the effective photocatalytic potential of NiO NPs. Insertion of GO into a TPU matrix provides strong H-bonding and improves the tensile strength of the membrane. Increasing the content of GO results in effective application up to a certain limit. Therefore, the novel combination of the TPU membrane proved efficient, in terms of stability and reusability. Only a 5% reduction in MO degradation was observed after 6 cycles of TNG-08 membrane reusability (Figure 8). After every trial, the membrane was collected, washed, and air-dried for the next reusability test. Besides, the TNG-08 membrane was used to find Ni leaching by ICP-OES. The treated sample contained 7 μg/L Ni concentration. The amount of Ni leaching in the treated sample was under the limit of WHO standards for drinking water (WHO/HEP/ECH/WSH/2021.6).

#### 3.2.5. Radical Scavenging Test

In photocatalytic degradation, reactive species, i.e., hydroxyl radicals (HO^•^); holes (h^+^); and electrons (e^−^) play key roles. To estimate the most effective radicals in this study, a radical trapping experiment was performed. Each scavenger, DMSO for ^•^OH, EDTA for holes, and K_2_Cr_2_O_7_ for electron trapping, was utilized in a 10 mM concentration under optimized reaction conditions [50]. The findings showed (Figure 9) that DMSO is the primary scavenger in the photocatalytic degradation process. Using a NiO-NH_2_-GO-4 (0.8% of membrane) photocatalytic membrane, the degradation values dropped dramatically from 94% percent to 30%, when DMSO was added. While, with the add-on of EDTA and K_2_Cr_2_O_7_, no effective decrease in dye degradation was examined as the degradation percentage changed from 94 to 73% and 80%, respectively. Therefore, hydroxyl radicals were found to be the key radical for MO dye degradation.

#### 3.2.6. Proposed Photocatalysis Reaction Mechanism

Considering the radical scavenging experiment, the estimated photocatalytic dye degradation mechanism by NiO/NH_2_-GO-based thermoplastic polyurethane membrane is presented as below in Figure 10: when the light of a particular wavelength (i.e., energy equivalent or greater than the energy bandgap of NPs) falls on an NiO/NH_2_-GO-based TPU membrane, the photoexcited electron of NiO NPs will make their way to the CB leaving holes in the VB. Integrating NiO NPs with aminated GO will improve the charge distribution ability of the composite membrane. The electron in the VB and the holes in the conduction band take part in the generation of a highly effective hydroxyl radical. Decomposition of hydrogen peroxide after the reaction with a superoxide radical also contributes to hydroxyl radical production. These radicals oxidatively degrade the pollutants into lower molecular weight products. The findings of radical scavenging experiments (Figure 9) support the above-described mechanism as hydroxyl radicals were found to be the key radical scavenger in the proposed photocatalysis degradation process (Figure 10).

#### 3.2.7. Effect of NiO/NH_2_-GO Concentration on TPU Membrane Performance

Figure 3 represents the SEM imaging of TPU-based membranes. Considering the scanning electron micrographs of TPU based membranes, it can be inferred that by increasing the concentration of NiO and NiO/NH_2_-GO the porous structure of the membrane decreases (Figure 3b,c). In Figure 3a, the concentration of NiO is the lowest, i.e., 0.2%, and the membrane is highly porous. With the insertion of a higher content of NiO/NH_2_-GO, the porosity decreases and surface roughness and density increase in the TPU based membranes. Therefore, successful cross linking between membranes and nanoparticles altered the filtration process.

The effect of NiO/NH_2_-GO concentration on pure water flux and MO rejection were analyzed under 4 bar pressure, and the results are presented in Figure 11. The graph presents a continuous decrease in pure water flux from 28.23 to 6.51 Lm^−2^h^−1^, with the increase in the concentration of NiO/NH_2_-GO in the TPU membrane. The decrease in pure water flux could be attributed to the formation of a new separation layer on the membrane and to membrane porosity.

The results of membrane’s porosity calculated by gravimetric analysis is presented in Table 2. A rapid decrease in porosity was observed when pristine TPU membranes were decorated with NiO NPs (i.e., membrane porosity decreased from 80.48% to 72.77%). However, the porosity of the NiO/NH_2_-GO membranes are comparable.

Besides, the dye rejection (%) increased from 42.4% to 92.6% by T-0 to TNG-10. This behavior is ascribed to the stronger electrostatic repulsion between the dye and TPU-based membranes. Concluding the water flux and dye rejection results, the optimum level of NiO/NH_2_-GO to the TPU membrane was selected as 0.8%, as further increasing the concentration could not provide appreciable results.

### 3.3. Kinetic Study

Herein, first- and second-order reaction kinetic models were chosen to analyze the dye degradation process. The expression for both kinetic models is presented in Equations (6) and (7), respectively. *K_1_* is the kinetic rate constant for the first-order model and *K_2_* is the kinetic rate constant for the second-order kinetic model, respectively.
(6)$$1\mathrm{st}\;\mathrm{order}\;\mathrm{kinetics}\;ln\frac{C_{o}}{C_{t}}=k_{1}.t$$
(7)$$2\mathrm{nd}\;\mathrm{order}\;\mathrm{kinetics}\;\frac{1}{C_{t}}-\frac{1}{C_{o}}=k_{2}.t$$

The regression analysis for both models is presented in Table 3 and Figure 11

As shown in Figure 12, the results revealed that the first- and second-order rate constant and regression co-efficient values were well suited to the first-order kinetic model as compared to the second-order kinetic model. According to the results, the best fitting with a perfect straight line had an R2 (goodness of fit) value of more than 0.97 for first-order reaction kinetic model. This indication supports that the degradation of MO follows a first-order kinetic model. For example, 0.97 means that at a specific reaction time (i.e., 180 min) 97% of the MO dye was degraded. An approximately 97% degradation efficiency of the MO dye was achieved under optimized conditions, so a first-order kinetic model is the best fit for the present study. The photocatalytic degradation reactions following first-order kinetic models involve rate determining steps, which depend on only one key reaction parameter (i.e., amount of photocatalyst used). The dye photocatalytic degradation process is a multi-factor dependent process but the rate determining step critically depends on the concentration of the photocatalyst.

Considering the results of the second-order kinetic model, the fitting is away from the straight line with a goodness of fit (R^2^) value of approximately 80%, which is considerably less as compared to the fitting of the first-order kinetic model. Similar results have also been previously reported .

## 4. Conclusions

Here, the novel combination of a TPU-NiO/NH_2_-GO composite membrane was explored to remediate the fouling problem associated with membrane technology in wastewater treatment. The membranes were fabricated using the phase inversion method, and they were investigated against modal anionic dye (i.e., methyl orange). The inclusion of NH_2_-GO not only enhanced the mechanical strength of as-prepared membranes but also greatly contributed toward the fostering of dye affinity toward their surfaces, owing to a point zero charge equilibrium along with the improved charge distribution capability of NiO results in a delayed electron–hole pairs recombination. Approximately 95% dye degradation of MO was achieved (under optimized conditions of solution pH = 4, H_2_O_2_ = 10 mM), within 180 min of reaction time under UV-254 nm irradiation. The proposed design of the composite TPU membrane also proved efficient in terms of stability and reusability. There was only a 5% reduction in the degradation potential of the membrane even after six cycles of reusability. In short, the proposed fabrication of the membrane can be effectively implemented in membrane wastewater treatment technology, minimizing the fouling problem associated with this technology.

## Figures and Tables

**Figure 1 membranes-12-00630-f001:**
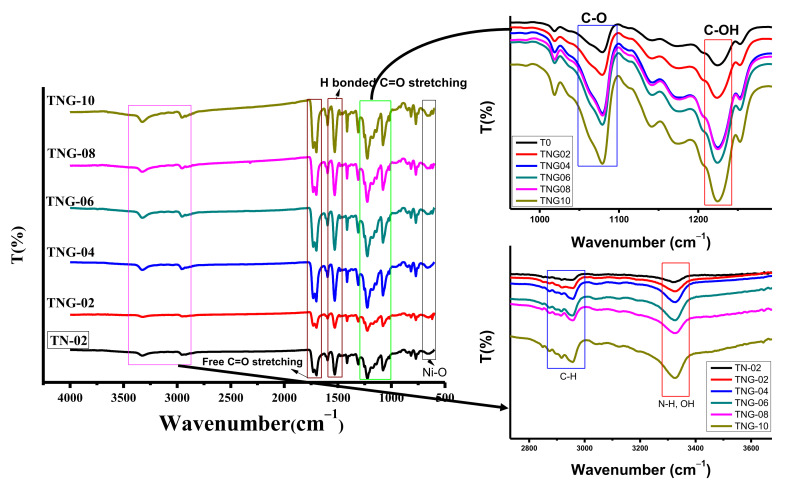
FTIR analysis of NiO/TPU and various ratio of NiO/NH_2_-GO based TPU membranes.

**Figure 2 membranes-12-00630-f002:**
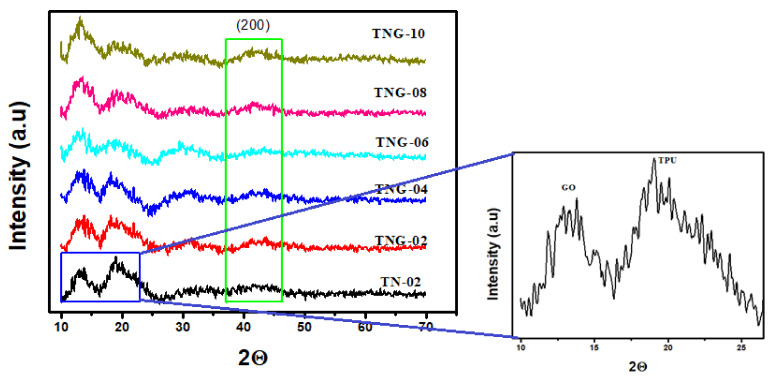
XRD pattern of NiO/TPU and various ratio of NiO/NH_2_-GO based TPU membranes.

**Figure 3 membranes-12-00630-f003:**
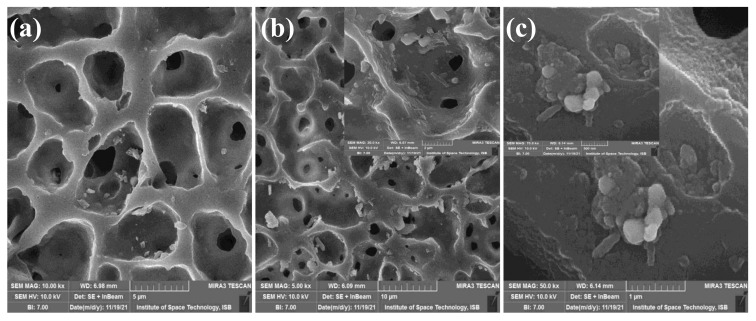
SEM imaging of (**a**) TN-02 (TPU-NiO), (**b**) TNG-02 (TPU-NiO/NH_2_-GO-1), and (**c**) TNG-10 (TPU-NiO/NH_2_-GO-5).

**Figure 4 membranes-12-00630-f004:**
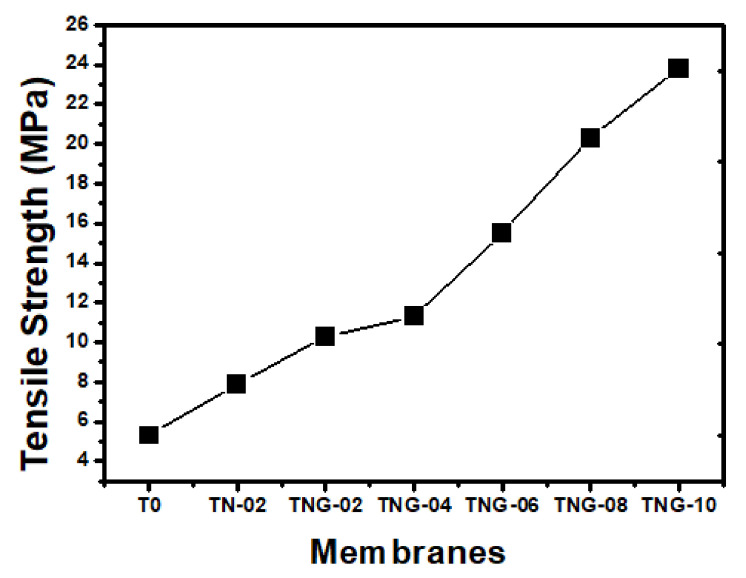
Comparison of tensile strength of the as-prepared membranes.

**Figure 5 membranes-12-00630-f005:**
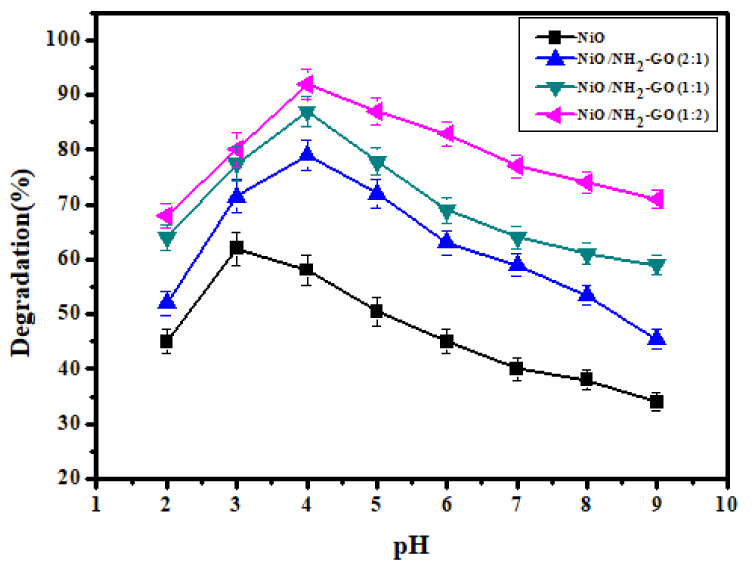
Effect of solution pH on photocatalytic degradation of MO using NiO and its composites with NH_2_-GO (reaction conditions = MO = 25 ppm, catalyst dose = 80 mg/100 mL, light source = UV 254 nm, H_2_O_2_ = 10 mM).

**Figure 6 membranes-12-00630-f006:**
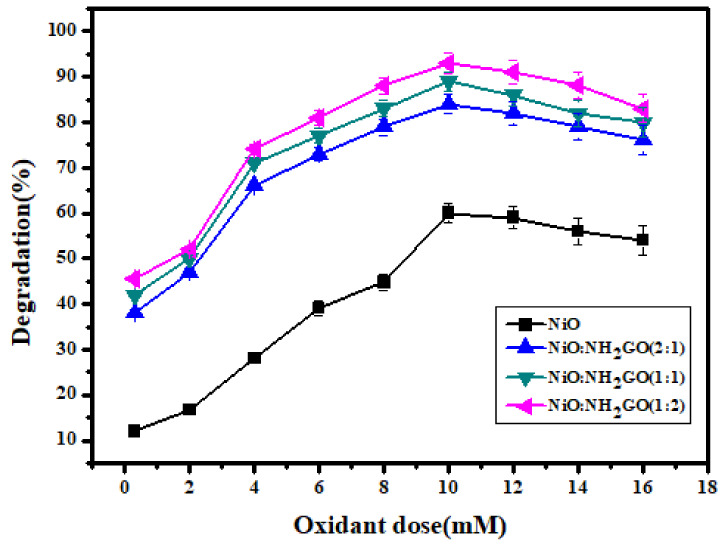
Effect of oxidant dose on photocatalytic degradation of MO using NiO and its composites with NH_2_-GO (reaction conditions = MO = 25 ppm, pH = 3 for NiO, NH_2_-GO, and NiO/NH_2_-GO92:1) and pH = 4 for other composites, light source = UV 254 nm, H_2_O_2_ = 10 mM).

**Figure 7 membranes-12-00630-f007:**
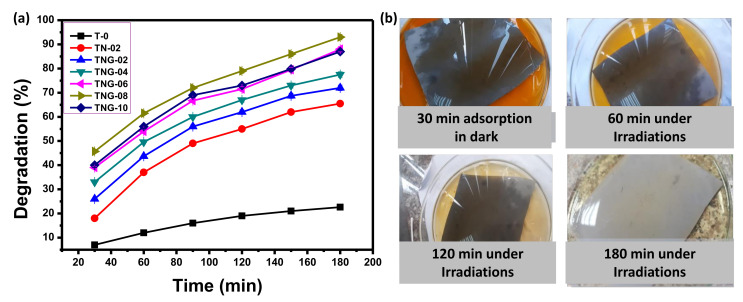
Effect of reaction time on photocatalytic degradation of MO (**a**) Using TPU and its composite membranes with NiO/NH_2_-GO (reaction conditions pH = 4, MO = 25 ppm, membrane area = 50 cm^2^, light source = UV 254 nm, and H_2_O_2_ = 10 mM), (**b**) visual description of TNG-08 working under optimized conditions.

**Figure 8 membranes-12-00630-f008:**
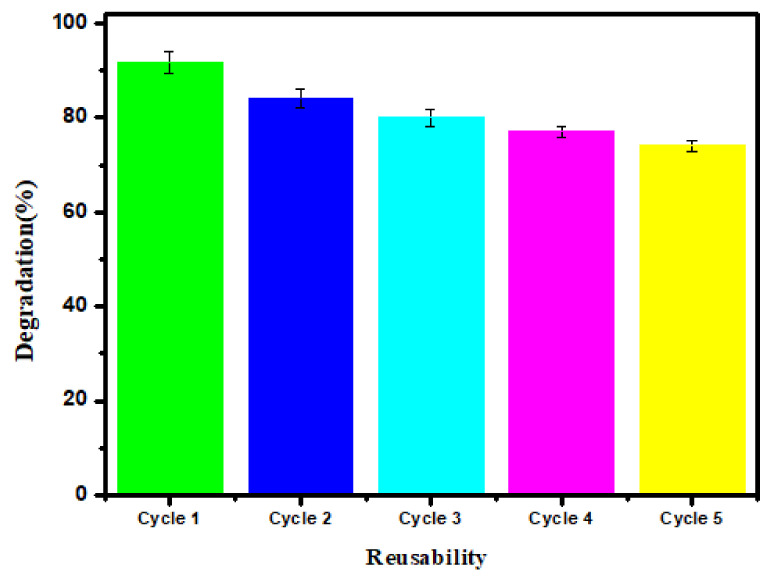
Stability analysis of TNG-08 under optimized conditions.

**Figure 9 membranes-12-00630-f009:**
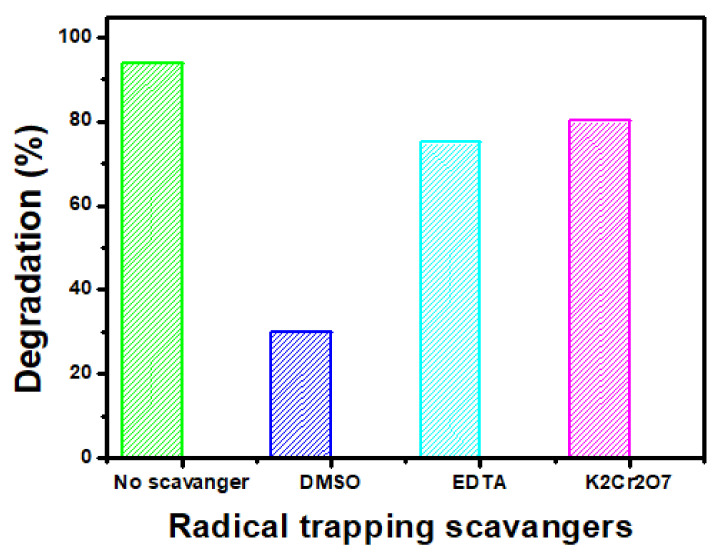
Estimation of effective radicals using key radical scavenger for TNG-08 under optimized reaction conditions.

**Figure 10 membranes-12-00630-f010:**
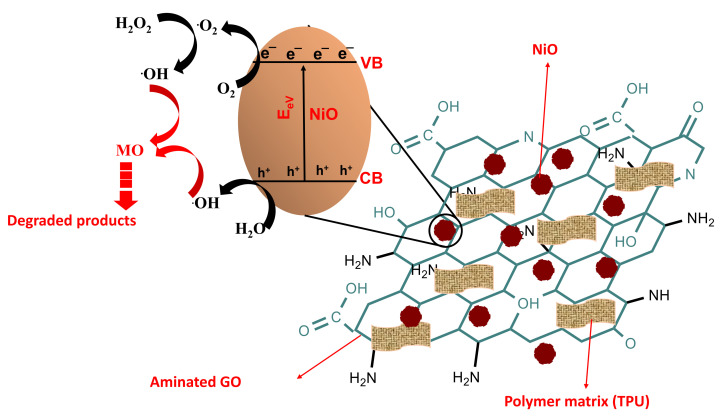
Proposed photocatalytic degradation mechanism using the results obtained from scavenging experiment.

**Figure 11 membranes-12-00630-f011:**
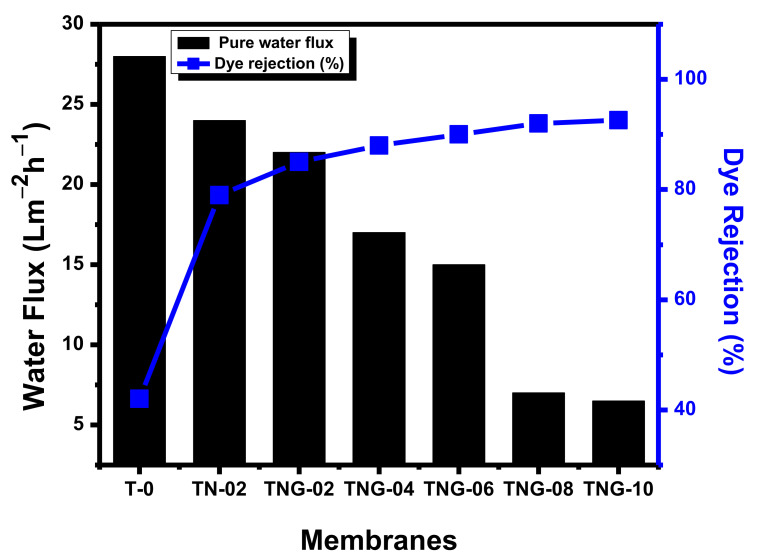
Pure water flux and MO rejection of TPU based membranes with different NiO/NH_2_-GO concentrations.

**Figure 12 membranes-12-00630-f012:**
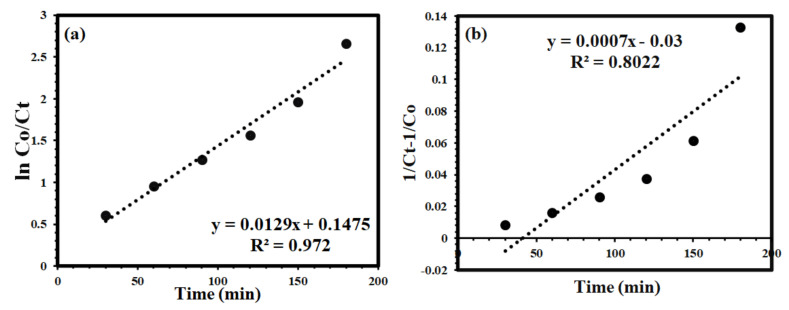
Kinetic models for the degradation of MO (**a**) 1st order, (**b**) 2nd order kinetic model using TNG-08 membrane.

**Table 1 membranes-12-00630-t001:** Composition of TPU-NiO/NH_2_-GO membranes.

Names	Membrane	TPU (wt.%)	DMF (%)	NiO-NH_2_-GO
T0	TPU	18	82	---
TN-02	TPU-NiO	18	81.8	0.2%
TNG-02	TPU-NiO/NH_2-_GO -1	18	81.8	0.2%
TNG-04	TPU-NiO/NH_2-_GO -2	18	81.6	0.4%
TNG-06	TPU-NiO-NH_2-_GO -3	18	81.4	0.6%
TNG-08	TPU-NiO-NH_2-_GO -4	18	81.2	0.8%
TNG-10	TPU-NiO-NH_2-_GO -5	18	81	1%

**Table 2 membranes-12-00630-t002:** Porosity (%) of TPU based NiO/NH_2_-GO membranes.

Sr#	Membranes	Porosity (%)
1	T-0	80.48
2	TN-02	72.77
3	TNG-02	73.17
4	TNG-04	71.53
5	TNG-06	70.42
6	TNG-08	68.94
7	TNG-10	69.32

**Table 3 membranes-12-00630-t003:** Values of co-efficient of correlation and rate constant of MO degradation using TNG-08.

Photo Catalytic Membrane	First Order		Second Order	
	R^2^	K_1_ (min^−1^)	R^2^	K_2_ (L µmol^−1^min^−1^)
TNG-08	0.97	0.012	0.80	0.0007

## Data Availability

Not applicable.

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
