# Peer review of "The Design of Ternary Composite Polyurethane Membranes with an Enhanced Photocatalytic Degradation Potential for the Removal of Anionic Dyes"

_membranes, 2022, doi:10.3390/membranes12060630_

Round 1

Reviewer 1 Report

The authors presented a composite membrane prepared by introducing NiO/NH2-GO into the TPU membrane matrix for enhanced photocatalytic performance. The photocatalytic degradation kinetic behavior and mechanism of the modified membrane are investigated. In general, the manuscript is fairly organized and of some interest for the development of fouling recovery of the membranes. However, some concerns should be addressed before publication.

(1) The basic comparisons of membrane characteristics need to be supplied, such as SEM, AFM....

(2) As a nanomaterials enhanced composite membrane, the leaching risk should be taken into the consideration;

(3) The overall distribution of nanocatalyst in the membrane matrix needs to be further presented in low magnification SEM to support the efficient utilization of nanocatalyst.

(4) More relevant works on composite membranes need to be cited in the manuscript to enrich the research background, such as DOI: 10.1016/j.memsci.2022.120493; DOI: 10.1016/j.seppur.2021.118567

Author Response

Attached file for reviewer 1

Reviewer 2 Report

Manuscript entitled “Design of ternary composite polyurethane membranes with enhanced photocatalytic degradation potential for removal of anionic dyes” submitted by Usman Zubair, Nimra Nadeem, Kainat Ghazal, Huda S. AlSalem, Mona S. Binkadem, Soha T. Al-Goul and Z.A. Rehan, can be considered for publication in Membranes Journal, after a serious major revision.

Here is a list of my specific comments:

  1. General comment: The utility of this study should be clearly highlighted in the manuscript. Pay attention on quantitative parameters of this process, and discuss its efficiency.
  2. Page 2, line 49: “The carcinogenic and mutagenic…”. Add here a reference.
  3. Page 2, line 57: “Numerous polymers are…”. The same observation as above.
  4. Page 2, line 83: “Various studies in the multidimensional field…”. Add here more references.
  5. Page 5, 2.4. Photocatalytic experimentation: The values of all experimental parameters (pH, adsorbent dose, volume, contact time, temperature, etc.) should be mentioned in this section.
  6. Page 5, line 202: “The absorbances of aliquots…”. The maximum wavelength should be mentioned here.
  7. Page 5, line 205: Replace “Here ?? is the initial concentration of dye and ?? is the dye concentration after a specific treatment time” with “where ?? is the initial concentration of dye and ?? is the dye concentration after a different treatment time”.
  8. Page 6, 3.1. Physicochemical Characterization: All results included in this section should be more detailed discussed.
  9. Page 9, 3.2.1. Effect of solution pH: At the end of this section, the optimal value of pH should be mentioned.
  10. Page 10, 3.2.2. Effect of oxidant dose: The same observation as above.
  11. Page 11, 3.2.3. Effect of reaction time: The results included in this section should be clearly presented and detailed discussed. The optimal value of this parameter should be also mentioned.
  12. Page 12, 3.2.4. Stability of composite in terms of reusability: What is this???
  13. Page 13, 3.2.6. Proposed photocatalysis reaction mechanism: The mechanism should be explained based on the experimental results. Therefore, this section should be clearly organized.
  14. Page 14, line 360: “It can be inferred that increasing the concentration…”. This observation should be supported by the experimental data.
  15. Page 15, 3.3. Kinetic study: The results included in this section should be clearly presented and detailed discussed.
  16. Page 15, 4. Conclusion: Include in this section the most important experimental results to highlight the importance of this study.

Author Response

Attached file 

Round 2

Reviewer 2 Report

Manuscript entitled “Design of ternary composite polyurethane membranes with enhanced photocatalytic degradation potential for removal of anionic dyes” submitted by Usman Zubair, Muhammad Zahid, Nimra Nadeem, Kainat Ghazal, Huda S. AlSalem, Mona S. Binkadem, Soha T. Al-Goul and Z.A. Rehan, can be considered for publication in Membranes Journal, after a minor revision.

Here is a list of my specific comments:

  1. Page 1, line 35: “The proposed composite membrane technology…”. This paragraph should be deleted.
  2. Page 2, line 75: “During water treatment…”. Add here a reference.
  3. Page 9, Figure 3: This figure should be reorganized. Each image should be noted and explained in the figure caption.
  4. Page 15, line 417: Replace “the SEM imaingimaging” with “the SEM images”.
  5. Page 16, line 454: “According to the results the best fitting…”. The results of kinetic modeling should be more detailed discussed.
